# Air Quality Monitoring Using Low-Cost Sensors in Urban Areas of Jodhpur, Rajasthan

**DOI:** 10.3390/ijerph21050623

**Published:** 2024-05-14

**Authors:** Ramesh Kumar Huda, Pankaj Kumar, Rajnish Gupta, Arun Kumar Sharma, G. S. Toteja, Bontha V. Babu

**Affiliations:** 1Indian Council of Medical Research, National Institute for Implementation Research on Non-Communicable Diseases, Jodhpur 342005, India; pankaj.k@dmrcjodhpur.nic.in (P.K.); babubontha.hq@icmr.gov.in (B.V.B.); 2Department of Community Medicine, University College of Medical Sciences, Delhi 110095, India; a.sharma@ucms.ac.in; 3Indian Institute of Technology, Jodhpur 342030, India; gstoteja@iitj.ac.in

**Keywords:** air pollution, ambient air quality, indoor air quality, PM_2.5_, microorganisms

## Abstract

Air pollution poses a significant health hazard in urban areas across the globe, with India being one of the most affected countries. This paper presents environmental monitoring study conducted in Jodhpur, Rajasthan, India, to assess air quality in diverse urban environments. The study involved continuous indoor and outdoor air quality monitoring, focusing on particulate matter (PM_2.5_) levels, bioaerosols, and associated meteorological parameters. Laser sensor-based low-cost air quality monitors were utilized to monitor air quality and Anderson 6-stage Cascade Impactor & Petri Dish methods for bioaerosol monitoring. The study revealed that PM_2.5_ levels were consistently high throughout the year, highlighting the severity of air pollution in the region. Notably, indoor PM_2.5_ levels were often higher than outdoor levels, challenging the common notion of staying indoors during peak pollution. The study explored the spatial and temporal diversity of air pollution across various land-use patterns within the city, emphasizing the need for tailored interventions in different urban areas. Additionally, bioaerosol assessments unveiled the presence of pathogenic organisms in indoor and outdoor environments, posing health risks to residents. These findings underscore the importance of addressing particulate matter and bioaerosols in air quality management strategies. Despite the study’s valuable insights, limitations, such as using low-cost air quality sensors and the need for long-term data collection, are acknowledged. Nevertheless, this research contributes to a better understanding of urban air quality dynamics and the importance of public awareness in mitigating the adverse effects of air pollution. In conclusion, this study underscores the urgent need for effective air quality management strategies in urban areas. The findings provide valuable insights for policymakers and researchers striving to address air pollution in rapidly urbanizing regions.

## 1. Introduction and Background Knowledge

Air pollution occurs when indoor and ambient air becomes contaminated by chemical, physical, or biological agents, resulting in a modification of the natural characteristics of the atmosphere. These agents, widely recognized as air pollutants, contribute to the alteration of air quality. Pollutants are solid, liquid, or gaseous substances found in the atmosphere at concentrations that can cause harm or have the potential to be detrimental to human beings, other organisms and the environment [1]. Particulate Matter (PM) is an intricate blend of solids and aerosols, comprising small droplets of liquid, fragments of dry solid, and solid cores coated with liquid. Based on size, PM is categorized mainly into PM_10_ and PM_2.5_ (diameter less than 2.5 μm) [2].

Bioaerosols are tiny airborne particles composed of living organisms and play a vital role in indoor air quality. These particles encompass bacteria, viruses, fungi, pollen, plant spores, dust mites, and other organic substances [3]. Bioaerosols can become airborne by breathing, talking, coughing, sneezing, and daily activities that disturb dust on indoor surfaces. It can contribute approximately 5–34% of particulate matter in indoor air. Occupational activities, such as waste sorting, composting, agricultural produce handling, food processing, and livestock handling, increase the risk of exposure to bioaerosols. The concentration and composition of bioaerosols in household environments vary significantly depending on air quality, relative humidity, temperature, etc. [4,5,6]. Exposure to high levels of bioaerosols can lead to various health issues, especially for individuals with respiratory conditions, allergies, or compromised immune systems, by activation of toll-like receptors (TLR-2 and TLR-4 HEK reporter cells) and inflammatory plasma biomarkers (IL-1Ra, IL-18, and TNFα) [7].

Brief exposures to PM_10_ (lasting up to 24 h) are mainly linked to the exacerbation of respiratory conditions such as asthma and chronic obstructive pulmonary disease (COPD), resulting in hospitalizations and visits to the emergency department [8]. Brief exposures to PM_2.5_ have been connected to premature mortality, heightened hospital admissions related to heart or lung issues, instances of acute and chronic bronchitis, asthma attacks, emergency room visits, respiratory symptoms, and days with limited physical activity. The consequences of extended exposure to PM_10_ are associated with conditions like COPD/Asthma, diabetes, and obesity. Prolonged exposure to PM_2.5_ has been correlated with premature mortality, especially in individuals with chronic heart or lung diseases, and diminished growth in lung function among children [9]. PM_2.5_ interacts with viral or bacterial aerosols, altering their infectivity. Therefore, it is perceivable that the presence of PM has considerably heightened the threat of microbial toxicity [10,11]. Consequently, the Central Pollution Control Board, India (CPCB) has implemented the National Ambient Air Quality Standards (NAAQS) guideline for PM_2.5_ as 60 µg m^−3^ (24 h) and 40 µg m^−3^ (annual) [12].

The study and analysis of household air quality have gained significant attention in recent years due to its direct impact on human health and well-being [13]. Understanding the distribution and composition of bioaerosols in residential settings is of utmost importance, as people spend a considerable amount of time indoors, especially in their homes. People residing in the same residence may experience shared exposures through direct contact with sources or indirectly through the contamination of the home environment [14]. According to IQAir’s World Air Quality Report 2022, of the 20 most polluted cities in central and south Asia, 15 are in India. Jodhpur ranks as the 54th most polluted city in the world. According to the same report, worldwide air pollution has cost seven million lives and $8 billion (USD) estimated daily economic cost (financial burden in 2021) [15]. The Central Pollution Control Board of India has one real-time monitoring site in Jodhpur that measures ambient air quality. Given that Jodhpur is located at the border of the desert, has several open-cast stone mines in its vicinity, and has a significant industrial hub in the western part of the city, air pollution is likely to be high, but the impact of this pollution on the lives of city citizens cannot be understood unless we assess the exposure at the individual level in the microenvironments of their daily living, be it at home, workplace, or commuting. Therefore, this study was conducted to determine the indoor and ambient air quality in different micro-environments of Jodhpur city of Rajasthan (India).

## 2. Methodology

### 2.1. Study Design

The study design was one of prospective longitudinal air quality monitoring.

### 2.2. Study Site/Areas

The study was conducted in the municipal area of Jodhpur city, which covered various residential premises, in slums, low, middle, and upper-income households, and commercial and industrial establishments. Jodhpur, Rajasthan, India, boasts a population density of around 1239 people per square kilometer. Its hot desert climate brings scorching summers with temperatures over 50 °C and mild winters averaging 28 °C. Humidity remains low, varying between 20% and 60%. Rainfall is sparse, primarily during the monsoon season from July to September.

### 2.3. Data Collection: The Following Data Were Collected

#### 2.3.1. Air Sampling (Air Quality Monitoring)

Portable electricity/battery-operated pre-calibrated air quality monitors (i.e., AirVisual Pro Indoor Sensor (IQAir)) were deployed in the selected households for one year (from June 2022 to May 2023). The sensors outdoors were adequately protected against heat, sunlight, rain, and wind, and we did not observe any malfunction of the sensors during the study period. In the present study, instruments were used with similar sensors in both indoor and outdoor environments. Low-cost sensors have lower accuracy and precision than EPA AQS monitors, which have the ability to provide useful information. Though low-cost sensors are dependent on environmental parameters, and their performance can degrade over time, they still offer sufficient accuracy for many applications, especially when used over extended periods. Variations across low-cost and precision monitors exist and, to minimize this, devices from one Original Equipment Manufacturer were used at all sites as reported in previous studies [16,17,18].

A laser particle scanner with a non-dispersive infrared sensor (NDIR) was installed for particulate matter for CO_2_. Deploying similar indoor and outdoor sensors offers consistency, versatility, and cost-effectiveness. Sensors assembled in indoor and outdoor settings provide cross-validation of data and uniformity in maintenance and calibration processes. Instrument calibration is necessary to improve the accuracy of low-cost sensors and is associated with machine learning-based calibration, like neural network ensembles and regression models, to advanced deep learning approaches [19,20,21]. This deployment covered two residential premises each in households in slums, low, middle, and upper-income colonies, and commercial and industrial areas. In each household one monitor was placed inside and another was placed outside the house. These air quality monitors collected real-time data on PM_1.0_, PM_2.5_, PM_10.0_, CO_2_ temperature, and relative humidity at a fixed interval of 5 min. The comparative campaign was also performed to assess the correlation between the instruments and inter-instrument deviations. The instrument was compared with the reference instruments operated by the manufacturing company, and as inter-instrument deviations were also observed by setting the instrument in standard conditions. The measured parameters for PM_2.5_ were 0–1000 μg/m^3^ ± 10 μg/m^3^/ or ±10%; temperature −40 to 90 °C/ −40 °F to 194 °F ± 2 degree C or F; humidity 0–100% RH ± 1%.

#### 2.3.2. Bioaerosol Sampling

Bioaerosol samples were collected from kitchen, bathroom, bedroom, living area and the outdoor environment surrounding the premises in each of the 12 households. A Petri-dish method was used for sampling from each of the above locations. In addition, The Anderson Cascade Impactor was employed for capturing airborne particulate matter, while Petri dishes containing Blood Agar and Chloramphenicol Rose Bengal Potato dextrose agar served as primary substrates for bacterial and fungal sample collection, respectively. Petri dishes were strategically, positioned at a height of 1 m, maintained at a distance of 1 m from obstructions, and sampled for 1 h. Subsequently, the collected Petri dishes were incubated for 48 or 72 h at 37 °C to facilitate bacterial growth for seven days. Post-incubation, colony counts were performed and expressed as Colony Forming Units (CFU/plate). The airborne particulate matter, like PM_2.5,_ was collected on filter paper by Anderson Cascade Impactor (8 h TWA). The PM_2.5_ concentration was calculated using the following formula:
$${PM}_{2.5}\;\mathrm{concentration}=\frac{Final\;weight-Initial\;weight\left(in\;µg\right)}{Volume\;of\;air\;sampled}$$
where total air sampled (in m^3^) = air sampling time in minute × air volume sampled (L/min).

Bioaerosol sampling by sedimentation method was performed using Petri plates containing nutritional medium without moisture. The plates were kept open for one hour and incubated at 37 ± 2 °C for 48 h for bacteria and 72 h for fungi. The count was expressed as total CFU/plate/hour (cfu/plate/h) [22].

### 2.4. Data Management and Analysis

All collected information was recorded digitally using an Excel sheet. Descriptive tables (Appendix A) were created showing all parameters’ daily and monthly averages. The air quality was compared indoors versus outdoors at each monitoring site and comparisons were drawn across all monitoring sites. The association of air quality with temperature, CO_2_, and humidity was also studied. Bioaerosol monitoring included identifying bacteria, viruses and fungi in the household air and described in terms of colony-forming units per plate of air and identification of pathogenic species.

Due to various technical reasons, the machines occasionally shut down, and the data were not captured during those short periods. These were treated as missing data points and no imputation was done to fill the missing points, as such incidents were insignificantly less in number. Similarly, we encountered a few faulty readings where the value of PM_2.5_ was below 20 µg/m^3^. Such data points were excluded from the analysis as extreme values that were unlikely to be correct.

All mean values were compared using independent *t*-test, and *p* value less than 0.05 was considered significant.

### 2.5. Ethics and Informed Consent

Ethical clearance was obtained from the Institute Ethics Committee. Written consent was obtained from the head of the household after providing a comprehensive description of the work. An information leaflet containing details of air quality monitoring and hazards of air pollution was shared with the participants.

## 3. Results

### 3.1. Overview

The real-time continuous air quality monitoring was done using factory-calibrated low-cost sensors in two households, each of six land-use patterns, with one monitor installed indoors and the other installed outdoors in the same household. Thus, data were available for 365 days from 24 devices with few missing data points because of power failure or malfunction. The rate of capture of the data was every five minutes. For analysis and brevity of presentation, the recorded data were converted into hourly, daily, weekly and monthly averages. The monthly average data are presented here (Appendix A). Daily and weekly data are provided in Appendix A. The data collection parameters were PM_1.0_, PM_2.5_, PM_10.0_, CO_2_, ambient temperature (°C), Air quality index and relative humidity. The monitoring device has an inbuilt correction factor to convert the raw value to a value corrected for temperature and humidity. We have used only adjusted values for analysis in this paper.

Figure 1 shows the monthly pattern of PM_2.5_ distribution as an average of all devices. We can see that the levels started rising from July itself gradually up to September, steeply up to November, steadily declining till May, and then rising again. The maximum average value peaked at 214.7 µg/m^3^ in November 2022 and the minimum average was 27.87 µg/m^3^ recorded in August 2022. There was a steep rise in October 2022, and high levels continued till February 2023. As the area falls in the northern hemisphere, the cooler months are October to February whereas in summer temperatures cross 40 °C in May and June. The average monthly temperature and humidity are shown in Figure 2 across all sites. July and August are rainy seasons, hence humidity is high, being a desert area, the rest of the months are relatively dry. The difference between indoor and outdoor humidity was non-significant (*p* = 0.6188) at 95% CI (−4.5496 to 6.9129).

### 3.2. Comparison of Indoor and Outdoor Values

Figure 3 shows a comparison of mean PM_2.5_ levels between indoor and outdoor areas over the study period. Throughout the year, the PM_2.5_ levels were higher inside the homes as compared to outside. The difference between the two was greater in the months of July, August, and January, but it was not statistically significant (*p* = 0.4532). Both indoor and outdoor concentrations were highest in slum areas, having an average of 106.8 µg/m^3^ and 87.8 µg/m^3^, respectively. In all types of localities, indoor PM_2.5_ concentration was higher than the outdoor concentration except in the HIG colony. But none of the differences were statistically significant (*p* > 0.05). The indoor-outdoor ratio of PM_2.5_ levels is depicted in Figure 4 and Figure 5. As we can see in Figure 4, the ratio has been above 1.0 throughout the year, and the difference between the indoor and outdoor month-wise ratio of PM_2.5_ level is statistically significant (*p* = 0.0072) with 95% CI(1.3961 to 7.0372). However, the high standard deviations of 30.77 (indoor) and 30.06 (outdoor) suggest significant variability, introducing uncertainty in these measurements. However, the maximum ratio was only 1.4 in August 2022. In Figure 5, the ratio was above 1.0 in all the localities except the HIG colony, and the difference between the indoor and outdoor locality-wise ratio of PM_2.5_ level is not statistically significant (*p* = 0.0612) with 95% CI(−0.0071 to 0.2137). The PM_2.5_ levels by type of locality is depicted in the Figure 6 and it shows that PM_2.5_ levels were higher inside the homes as compared to outside in all localities (except the HIG colony).

The temperature and humidity levels are depicted in Figure 7 and Figure 8, respectively. The difference between indoor and outdoor temperature was statistically significant (*p* < 0.0001) with 95% CI (−6.4068 to −4.2499). Outdoor temperatures were higher than indoors in industrial, commercial, and MIG colonies and were reversed in LIG and HIG colonies. However, the differences were marginal and not at all significant. It was equal in slum areas. Humidity levels were also very similar in indoor and outdoor areas in all types of colonies. Appendix A presents the monthly average Outdoor Air Quality Index (AQI US) data for other areas of Jodhpur over the same year in certain regions, such as the Industrial and Commercial Areas, the average indoor AQI tended to be higher than the outdoor AQI. This suggests that indoor sources of pollution, such as emissions from industrial processes or indoor activities, may contribute significantly to air quality degradation within these areas. Conversely, in the Slum Area, the average indoor AQI was consistently lower than the outdoor AQI. This indicates that outdoor pollution contributes more significantly to air quality issues in these regions, while indoor air quality is comparatively better. The Middle-Income Area showed a closer alignment between indoor and outdoor AQI values, suggesting that indoor and outdoor sources contribute to air pollution in this region. Overall, the data highlight the varying indoor and outdoor air quality levels across different areas of Jodhpur city. Appendix A shows that indoor CO_2_ levels tended to be higher than outdoor CO_2_ levels across all sites of Jodhpur City, indicating potential indoor air quality concerns in various households.

### 3.3. Temporal Comparisons in Each Type of Colonies

Figure 9 shows that indoor areas were most polluted in houses in the slum colony. But in July and August, LIG colony and industrial areas had more polluted households. Similarly, Figure 10 shows that PM_2.5_ levels in the outdoor areas were also high in the slum area but there was wide variation between LIG, MIG and HIG colonies with changing levels, particularly in the winter months. The difference between indoor and outdoor PM_2.5_ levels is non-significant (*p* = 0.0735) with 95% CI (−1.0139 to 15.7106).

### 3.4. Atmospheric Determinants of Air Quality

The correlations among temperature, humidity, and PM_2.5_ are depicted in Table 1. Although the correlation between PM_2.5_ and temperature appeared good, it was not statistically significant (*p* value = 0.18). The correlation was examined for the entire duration of the study and applied on monthly data. Moreover, both the correlations between PM_2.5_ and humidity and temperature and humidity were observed to be weak. Possibly there are several other variables that confound the correlation between weather parameters and PM_2.5_, though these were not examined, hence it is difficult to explain the weak correlation.

### 3.5. Bioaerosols

The presence of microbes in different urban areas was recorded using the Petri dish method in each premise’s kitchen, bathroom, bedroom, living area, and outdoor area. Bioaerosol sampling was conducted in November 2022, February, May, and July 2023 to capture seasonal variations.

As per bioaerosol monitoring performed in November 2022, the highest bacterial counts were found in the living area of industrial localities and high-income colonies, followed by the outdoor area in slum areas and bedrooms in slum housing. The highest fungal counts were observed in the kitchen and outdoor areas of the commercial colony. The lowest bacterial and fungal counts were observed in the low-income colony (Appendix A). Based on microscopic as well as biochemical examination of bacterial isolates, different bacteria were identified, including coagulase-positive *Staphylococcus*, coagulase-negative *Streptococci*, coagulase-negative *Cocci*, coagulase-negative *Staphylococcus*, and Gram-negative *Diplo* and *Streptobacilli*. Fungal examination revealed the presence of *Aspergillus niger* and *Penicillium* spp. in household air samples.

In February 2023, bacterial colony-forming unit counts (CFU/plate) were highest in the industrial area outdoor area (n = 576), followed by the low-income colony bathroom area (n = 488) and commercial area kitchen area (n = 472). The lowest bacterial concentration was found in the high-income colony location (bedroom, bathroom, and living area, n = 57, 68, and 76, respectively). The highest fungal counts were observed at the low-income colony location (outdoor, n = 128; bathroom, n = 89; bedroom, n = 83), slum area outdoor (n = 81), low-income colony kitchen (n = 76), and slum area living area (n = 61). The lowest fungal counts were observed in the middle-income colony (living area, n = 9, and kitchen, n = 10) (Appendix A). Microscopic as well as biochemical tests revealed the presence of bacteria, including coagulase-positive *Staphylococcus*, Gram-positive *Cocci*, Gram-positive *Streptococci*, coagulase-negative *Staphylococcus*, and Gram-negative bacilli. Fungal examination revealed the presence of *Aspergillus* spp., including *A. niger*, *Penicillium* spp., etc., in household air samples.

In May 2023, bacterial colony-forming unit counts (CFU/plate) were highest at the middle-income colony location (bathroom, n = 362; outdoor, n = 348; bedroom, n = 308; and kitchen, n = 292). The presence of bacteria was also higher in the kitchen area, bathroom, outdoor, and living room of slum areas (kitchen, n = 324; bathroom, 275; outdoor, n = 264; and living area, n = 208). The lowest bacterial counts were present in the industrial area bedroom (n = 32), bathroom (n = 51), and kitchen (n = 63) area. The highest fungal counts were observed at the slum areas location (living area, n = 27; outdoor, n = 26), high-income colony (outdoor, n = 26), commercial area (outdoor, n = 24; living area, n = 19), the outdoor area of the low-income colony, and middle-income colony (n = 15 and n = 14, respectively). No fungal isolate was collected from the bedroom area of the industrial area (Appendix A). Microscopic and biochemical tests revealed the presence of bacteria, including coagulase-positive *Staphylococcus*, Gram-positive *Cocci*, coagulase-negative *Streptococci*, and coagulase-negative *Staphylococcus*. Fungal examination revealed the presence of *Aspergillus* spp., including *A. niger*, *Penicillium* spp., etc., in household air samples.

As per the bioaerosol monitoring performed in July 2023, the majority of locations had bacterial colonies ‘too numerous to count (TNTC),’ except at the low-income colony bathroom area (n = 216), kitchen area (n = 160), middle-income colony living area (n = 192), and commercial area bedroom area (n = 186). The RBPDA plating for fungal culture revealed the CFU/plate at the commercial area living and outdoor areas (n = 69, n = 65, respectively), followed by the slum area outdoor area (n = 57), bathroom area (n = 53), and living area (n = 36). The minimum fungal CFU counts were observed in the middle-income colony bathroom area (n = 5) (Appendix A). Microscopic and biochemical examination revealed coagulase-negative *Streptococci*, coagulase-positive *Diplococci*, and coagulase-negative *Streptococci*. Fungal isolates observed were *Aspergillus niger* and *Penicillium* spp.

## 4. Discussion

Air pollution is a significant health hazard in urban areas worldwide. Of the hundred most polluted cities in the world, India ranks 64. Jodhpur ranks as 56th most polluted city globally and 43rd most polluted city in India in 2022 [23]. Adequate evidence exists for suspended particulate matter being a risk factor for respiratory illnesses, like bronchial asthma, Chronic Obstructive Pulmonary Disease, and allergic rhinitis [24,25,26]. The role of airborne particulate matter in adverse pregnancy outcomes, increased risk of mortality due to cardiovascular diseases and diabetes mellitus is also evidenced in recent studies [27,28,29,30]. The possible role of air pollution in neurological illnesses is also being explored scientifically [31]. According to the National Air Quality Standards of India, the permissible limit of PM_2.5_ in ambient air at 24 h average is 60 µg/m^3^ [32]. At the same time, WHO prescribes a 24-h average value of 15 µg/m^3^ and an annual mean at 5 µg/m^3^. According to WHO, in 2019, 99% of the world’s population lived in areas that did not meet the prescribed levels of PM_2.5_ in ambient air [33]. PM_2.5_-bound toxic contaminants are associated with multiple chronic health problems in human. Estimates have also shown that about 7 million people worldwide are estimated to lose their lives prematurely due to air pollution [34]. Such is the catastrophic effect of air pollution on human lives, but despite this, neither is air quality monitored effectively in most of the cities in our country, nor are any concerted efforts being made to reduce air pollution. Jodhpur, the second largest city of Rajasthan after Jaipur, had only one ambient air quality monitoring site and agencies are yet to monitor indoor air quality. Hence, the true amount of exposure is unknown, as we do not measure the values in indoor environments where people spend most of their time. For this reason, in this study, we monitored the air quality in indoor and ambient air simultaneously so that the extent of exposure can be truly estimated.

### 4.1. PM_2.5_ Levels

The average monthly PM_2.5_ levels ranged between 50 and 100 µg/m^3^, but peak levels touched 214 µg/m^3^ and minimum values dipped to 46 µg/m^3^. The concentration of PM_2.5_ measured by both by Anderson cascade impactor and by low-cost sensor resulted in variations which can be attributed to several important factors, mainly measurement principle, sampling duration and frequency (the low-cost sensor recorded continuous data for 365 days at 5 min intervals while the cascade impactor was run for 8 h TWA), sensitivity and detection limits, environmental factors, and data processing interpretation, each of which affects the measurement process differently.

The next important observation was that indoor PM_2.5_ levels were higher than the outdoor levels throughout the year when the average of all sites was compared on a monthly basis (Figure 4), thus letting us infer that in Jodhpur the message of staying indoors during peak pollution periods may be counterproductive. This is also evidenced by the I/O ratio of PM_2.5_ being always greater than 1, except in the high-income colony and locality where the monitoring was being done, and it remained the same throughout the monitoring period of one year. The most polluted indoor air with PM_2.5_ was found in the slum area, followed by the industrial and commercial areas. The houses monitored in the HIG colony had the lowest annual average PM_2.5_ levels and this was the only area where indoor PM_2.5_ levels were better than outdoor ones. The results of the present study are similar to the studies reported previously [35,36]. We also observed differences in ambient temperature across the colonies. Indoor areas were cooler except in low and high-income housing colonies. Thus, in high-income housing areas, the indoors were less polluted and had low temperatures because of the use of air conditioners. Though relative humidity was higher in indoor monitoring sites than in outdoor sites, the difference was marginal. It can be concluded that better air quality prevailed in the high-income houses, thus it is possible to bring down pollution levels by suitable interventions. Similar results were observed by Barrington-Leigh et al. 2019, where air conditioning was responsible for less indoor pollution and low temperature [37]. Understandably, air conditioners cannot be afforded by lower-income households. Still, other cooling techniques, like desert coolers, proper ventilation or simple humidification of the houses, can control the level of particulate matter in the air. Temporally speaking, throughout the year, the best air quality was seen in the high-income colony, and it will be prudent to study the factors that contributed to keeping these pollutant levels low. However, it should also be noted that the outdoor air quality was not the best in the high-income colony. In winter (peak pollution period), the best PM_2.5_ levels were seen in the low-income colony, but in summer, it was better in the middle-income colony. This heterogeneity needs to be explored further to determine the factors responsible for it. We assessed the correlation between PM_2.5_, temperature, and humidity. The correlation coefficients indicated a moderate positive correlation between PM_2.5_ and temperature (Pearson’s correlation coefficient = 0.62) and a weak correlation with relative humidity (R = 0.231). However, there was no correlation between temperature and humidity (Table 1).

### 4.2. Bioaerosols

The microbial counts showed wide variability. The living areas in all housing types had maximum CFU counts for bacteria. This is most likely because this is the room where people enter from outside and are also likely to wear footwear, bringing organisms inside. Association between home characteristics, footwear and concentrations of bacteria, fungi and endotoxins were reported by Sousa, 2022 [38]. It is also the most exposed part of the home to the outer atmosphere. Hence microbial counts are high. The highest count was in the living area in the industrial region (428 CFU/plate) followed by the slum area. Compared to other colonies, the outdoor bacterial count was highest in the industrial area (576 CFU/plate). Garbage collection and cleanliness of the streets are least common in industrial areas as well as slum areas, therefore, predictably the CFU count was highest in this part. But otherwise, there was no definite pattern visible and it seems from the data that land use pattern and socio-economic status are unrelated to the distribution of colony counts. Microscopy and biochemical analysis showed that several pathogenic organisms were present in the growth on the Petri dish. This included staphylococcus species, streptococcus, gram-negative diplo cocci and streptobacilli, hence posing a risk of infection among the residents. The fungal growth was also present at all sites and in all four rounds except in the houses in the industrial area.

## 5. Limitations

In this study, we used low-cost, laser-based air quality sensors. The accuracy of the sensors is low compared to high-end sensors; nevertheless, in the literature, various researchers advocate the use of such sensor-based monitors. One year’s data are insufficient to comment upon time trends and long-term trends in pollution levels. Bioaerosol monitoring was less frequent due to logistical constraints.

## 6. Conclusions

Air quality monitoring conducted indoors and outdoors gave a quantitative picture of the extent of air pollution in households in different land use patterns. The spatial as well as temporal diversity of the extent of air pollution in the microenvironments within a city was successfully captured in this study. Thus, household air quality monitoring can be performed to obtain a real-time assessment of air quality in several diverse microenvironments that prevail in a city. Threat to human health is real from the air pollutants, as we observed that pollutant levels were far beyond permissible limits throughout the year in every part of the city. Hence, this study underscores the urgent need for effective air quality management strategies in urban areas. The findings provide valuable insights for policymakers and researchers striving to address air pollution in rapidly urbanizing regions.

## Figures and Tables

**Figure 1 ijerph-21-00623-f001:**
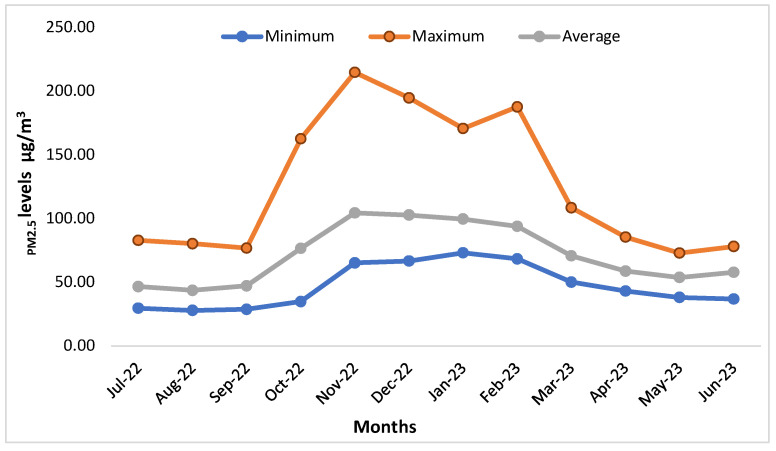
Monthly average PM_2.5_ levels (MinPM_2.5_= Minimum value, MaxPM_2.5_ = Maximum value).

**Figure 2 ijerph-21-00623-f002:**
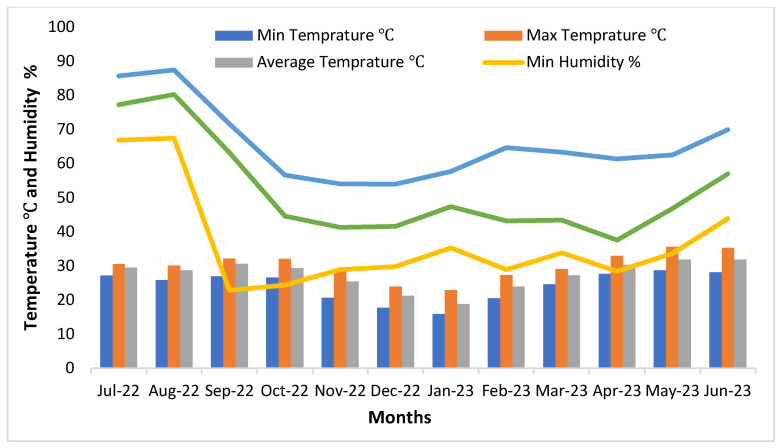
Monthly average temperature and humidity across all sites.

**Figure 3 ijerph-21-00623-f003:**
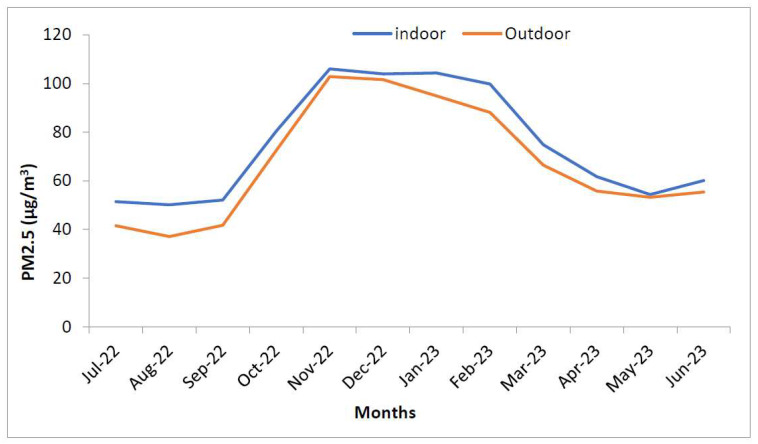
Comparison of PM_2.5_ levels of indoor and outdoor locations across the year.

**Figure 4 ijerph-21-00623-f004:**
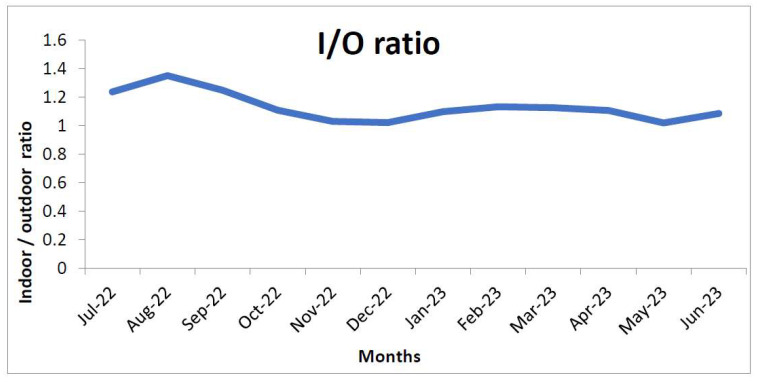
The month-wise ratio of PM_2.5_ levels (indoor and outdoor).

**Figure 5 ijerph-21-00623-f005:**
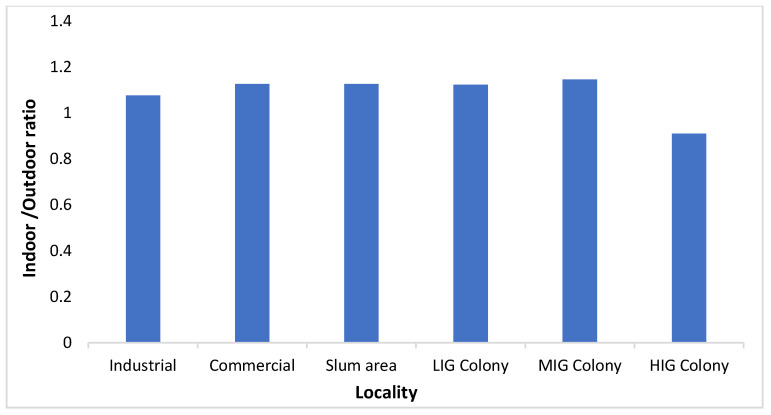
The locality-wise ratio of PM_2.5_ levels of indoors and outdoors (I/O ratio).

**Figure 6 ijerph-21-00623-f006:**
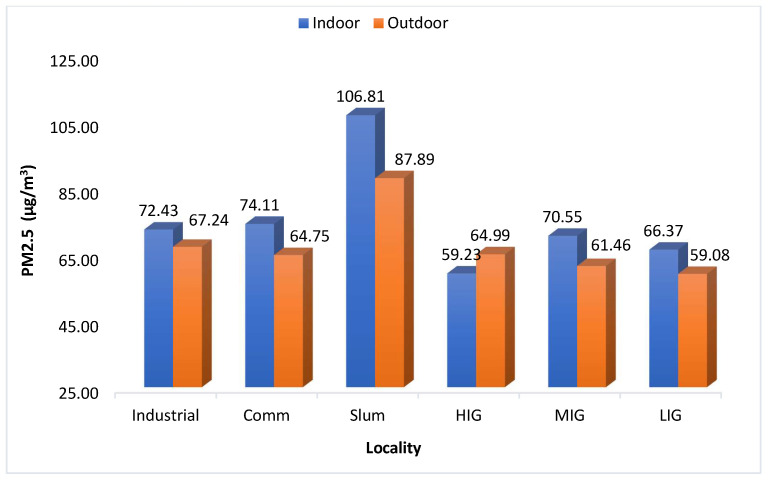
Locality-wise comparison of Indoor and Outdoor PM_2.5_ concentration (365 days Avg.).

**Figure 7 ijerph-21-00623-f007:**
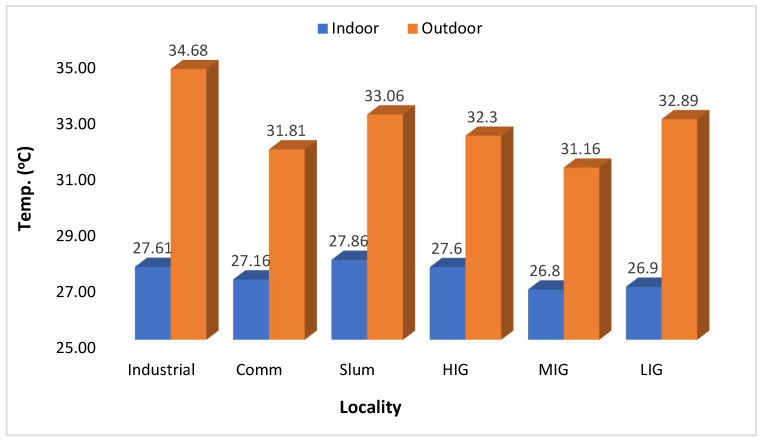
Locality-wise comparison of indoor and outdoor atmospheric temperature (365 days Avg.).

**Figure 8 ijerph-21-00623-f008:**
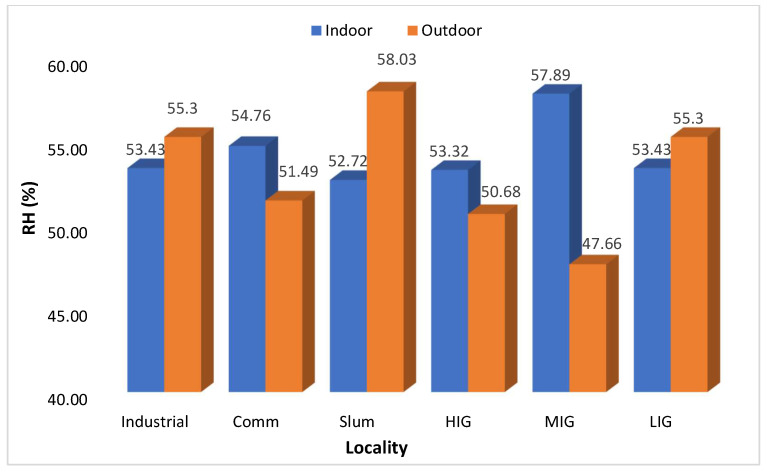
Locality-wise comparison of indoor and outdoor atmospheric relative humidity 365 days Avg.).

**Figure 9 ijerph-21-00623-f009:**
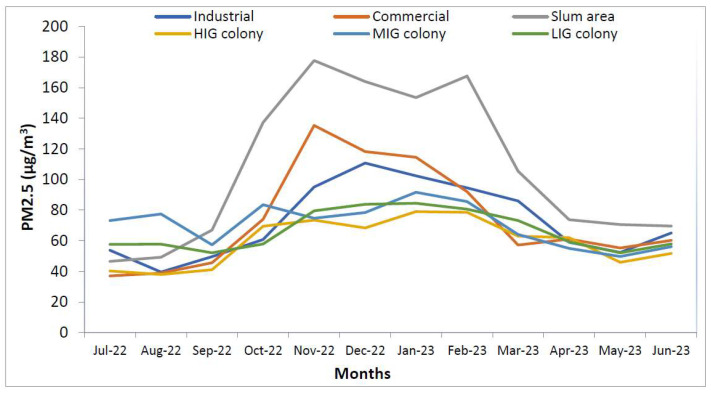
Temporal distribution of PM_2.5_ levels in indoor areas of all colonies.

**Figure 10 ijerph-21-00623-f010:**
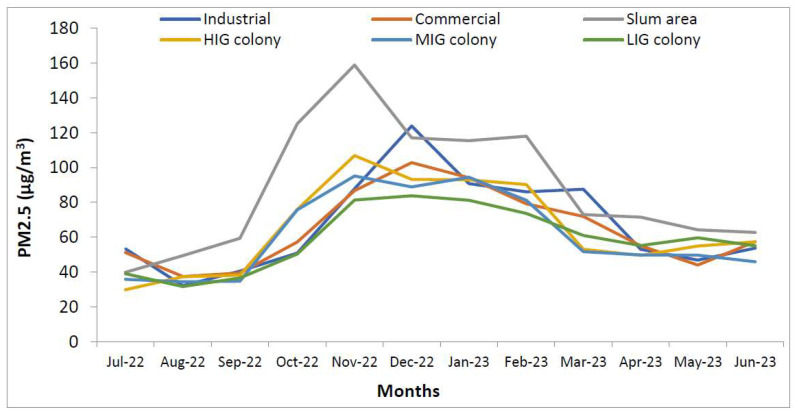
Temporal distribution of PM_2.5_ levels in outdoor areas of all colonies.

**Table 1 ijerph-21-00623-t001:** Correlation between temperature, humidity and PM_2.5_.

	Temperature	Humidity
PM_2.5_	0.62	0.231
Temperature	NA	0.09

## Data Availability

The data presented in this study are available on request from the corresponding author due to ethical reasons.

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
