# Peer review of "Air Quality Monitoring Using Low-Cost Sensors in Urban Areas of Jodhpur, Rajasthan"

_ijerph, 2024, doi:10.3390/ijerph21050623_

Round 1
Reviewer 1 Report
Comments and Suggestions for Authors
Comments
The study provides valuable insights and produces data to evaluate air pollution levels indoors and outdoors in India. It can be very useful to policy makers to implement actions to improve air quality.
Nevertheless, there are certain issues that could be resolved to increase the quality of the manuscript.
Comments for the whole study:
1) The study is referring to understanding the population exposure, but only monthly and daily concentrations are reported and there is no correlation to the time spent in each indoor microenvironment or in the outdoor environment. Maybe the authors could state that this study could be used to further evaluate exposure to PM.
2) Section 3.4 is very weak. Further attempt to depict possible correlations must be done.
3) In the Discussion section there are no references to compare the findings with other studies in India or in areas with similar environmental and socioeconomical characteristics. In general Discussion seems very weak and mostly used to repeat the findings of the Results.
Specific comments:
1) Section 2.3: The reference site could be used to calibrate the portable instruments used un the study. If this is not possible, some references could be added to support the performance of the sensors in various environments.
2) Sections 3.1 and 3.5: Maybe data could be grouped more and presented in brief. Tables 1-5 and 7-9 could be used as supplementary material. Furthermore, more statistics such as median values, standard deviations and quartiles could be presented.
3) In table 5, the scale of the AQI index could be mentioned.
4) Table 10 is not in the right place. Should be closer to section 3.4.
Comments on the Quality of English LanguageEnglish language used is adequate and no issues are reported.
Author Response
Respected Sir/Madam,
Following your suggestions and comments, I have revised the manuscript, and the same has been highlighted in light blue in the revised manuscript. A response to each comment is enclosed for your consideration.
Thank you.
Sincerely,

Reviewer 2 Report
Comments and Suggestions for Authors
The paper "Air Quality Monitoring Using Low-cost Sensors in Urban Areas of Jodhpur, Rajasthan" describes air-quality measurements in Jodhpur. PM levels and environmental parameters such as T, RH and biosol types and concentrations were measured and evaluated.
Low-cost aerosol monitors were used for PM1.0, PM2.5, PM10.0, CO2, T, and RH measurements. For bioaerosol sampling, the particles were collected on filter paper in an impactor and then counted. The air quality data was sampled for 12 months, indoors and outdoors, and in several typical locations. The microbial counting was performed four times during the same one-year period. The authors show a basic statistical analysis of the data (mostly monthly average, min, and max).
The research topic is interesting and addresses an actual applicative problem; however, the Paper itself has several flaws.
The low-cost sensors can be very inaccurate, dependent on environmental parameters, and their performance can degrade over time, especially outdoors. The authors acknowledge the limitations of the low-cost sensors in the section "5. Limitations" but refer to the literature as the advocacy of using such sensors in research. As low-cost optical sensors can have huge errors, the authors must clearly explain:
- Which sensors were used exactly in the indoor and outdoor placement? It seems now that sensors designed for indoor use were also used outdoors.
- How were they calibrated?
- Was there a comparative campaign in which the correlation between the instruments was assessed and inter-instrument deviations were established?
Such backup is necessary for the collected numbers to be compared to any legislative limits; otherwise, no claims about air quality can be made.
There is no uncertainty assessment of the instrumental data. Unless the instruments were carefully calibrated and the above-mentioned statistical analysis performed, the errors could be even higher than 50%. The monthly averages are given in two decimals (while it is not clear whether the PM instruments were even 10% within the given value). Also, temperature and humidity averages are given to two decimals; such precision should be explained.
The Paper needs improved statistical handling. Only the monthly average and maximum and minimum values are given in the present version. Many more statistical parameters (e.g., standard deviation), hourly, daily, and monthly correlations between sites, instruments, etc., would be interesting in such a study.
The Authors mention several correlation values but omit the time frame of those calculations. The correlations can differ when observing hourly, daily, weekly, and monthly scales. The Authors also do not describe the weekly and daily behaviour of the variables to observe daily and weekly cycles.
Some claims are contradictory, such as a "strong negative correlation between T and RH," while the correlation coefficient in Table 10 shows 0.09, which means almost no correlation and is positive.
The authors use the term "statistically significant" several times, giving the value of p (sometimes P) without ever naming this parameter.
It would be beneficial to validate the measured data, for example, by comparing the T and RH from low-cost sensors to other sources (external meteorological data available). The Authors could also compare the PM2.5 data obtained by weighting the PM matter (see section 2.3.2) and PM2.5 data from the optical sensor to self-evaluate.
Several tables and Figures lack units. The capital M is used for Meter, which is wrong.
The Paper contains many numerical tables and figures, with graphs that do not increase its scientific value. The impression is that many of the huge graphs are there just to fill up the Article to 20 pages without giving any deep insight into the data. Most of the Tables and Figures should be moved to the Supplementary data and only graphs that give relevant insight and are comprehensively discussed in the text should be kept in the Paper.
The general impression of this work is that the research has produced a large amount of very valuable and good data (one year, 24 instruments, 5-minute resolution). However, there is a lot to be improved on the data analysis side and the data interpretation. The Article's main point (PM concentrations above the legislation limits) should be backed up by providing more data about the instruments.
Comments on the Quality of English LanguageI have no comments on the language. There are some typos, like Fig6 (Oitdoor)...
Author Response

(The authors gave the same response as above.)

Round 2
Reviewer 2 Report
Comments and Suggestions for Authors
The authors have included most of the detailed comments, but they should have also majorly revised the Paper based on more general comments. This still needs to be done.
All the details about the calibration of the low-cost sensors provided in Replies 1, 2, 3, and 4 of the Response to Reviewer Comments should be included in the Paper as well (included in Section 2.3.1).
The uncertainty must be appropriately calculated and given in the Paper for all the presented data! This is necessary to explain if the I/O ratios in Figures 5 and 6 are statistically significantly deviant from 1.0 or if it is a normal variation due to the uncertainty of the values and if there is no difference in Indoor and Outdoor air quality. In other words, it should be shown that I/O ratios are different than 1.0 above the uncertainty doubt (or not).
The same is true for Figures 10 and 11: uncertainties are needed to assess whether the differences between curves are significant.
Calibration and known uncertainties are necessary to compare the measurements with the regulatory values and give statements like "exceeds permissible limits"; otherwise, such statements must be removed.
The Authors did not answer my comment about comparing the PM2.5 data obtained by weighting the PM matter (see section 2.3.2) with PM2.5 data from the optical sensor to self-evaluate.
Most graphs still need to be revised to bring new scientific contributions. Figures 2 and 3 (T and RH) do not describe air quality (the Paper's main topic) but tell the general climate conditions in the area, so they should be moved to the Supplementary material. There is no new scientific value in those graphs. For example, googling for "Jodhpur climate" and looking at the first hit https://en.climate-data.org/asia/india/rajasthan/jodhpur-2848/, one gets the same graph of temperatures: from a little above 15°C in winter months, to about 35°C in summer. Even the second temperature peak in September can be seen, exactly the same as in Figure 2 of the Article. Alternatively, those graphs can be compressed into one Figure; now, they are taking too much space and bring little value.
The Authors should explain why their study regarding outdoor PM is better than the data already available on the internet, e.g. on https://www.aqi.in/dashboard/india/rajasthan/jodhpur
where the data can be seen and observed on many different time scales in a user-friendly manner.
Spatial data and indoor/outdoor ratios are more innovative and of higher scientific value; however, graphs are again too basic. Graphs focusing on variations should not always have an origin at 0 (for example, Figures 5, 6, 7, 9,...). Imagine how Figure 8 would look if scaled from 0. In the present version it is an example of proper scaling, starting from 26°C. Some graphs are still missing the labels (Figure 1).
The lack of correlation between T and RH (and other parameters) is understandable on the monthly level. Those two parameters are correlated when the absolute amount of water vapour in the air remains constant, which happens on a short time scale only (hours, days). My guess is all those correlations would be much stronger if calculated on shorter timescales.
Additionally, NAAQS defines PM guidelines in terms of 24-hour average and annual average. None of them was calculated in this study, which should be commented on.
As the authors do not wish to expand the data analysis on the shorter timescales (daily, weekly) in this Paper and opt for another one—see comment 6—they should remove the label "comprehensive" for this study. The current comparison and analysis of the monthly averages is no more than an elementary study.
This is one of the main problems I see in this article. The data analysis is too basic, leading to unfounded conclusions, so I do not recommend that the Paper be published in the present form.
Author Response

(The authors gave the same response as above.)
